# Improvement of Misleading and Fake News Classification for Flective Languages by Morphological Group Analysis

**Jozef Kapusta ***  **and Juraj Obonya**

Department of Informatics, Constantine the Philosopher University in Nitra, Nitra SK-94974, Slovakia;
juraj.obonya@ukf.sk
* Correspondence: jkapusta@ukf.sk

**Abstract:** Due to the constantly evolving social media and different types of sources of information, we are facing different fake news and different types of misinformation. Currently, we are working on a project to identify applicable methods for identifying fake news for floating language types. We explored different approaches to detect fake news in the presented research, which are based on morphological analysis. This is one of the basic components of natural language processing. The aim of the article is to find out whether it is possible to improve the methods of dataset preparation based on morphological analysis. We collected our own and unique dataset, which consisted of articles from verified publishers and articles from news portals that are known as the publishers of fake and misleading news. Articles were in the Slovak language, which belongs to the floating types of languages. We explored different approaches in this article to the dataset preparation based on morphological analysis. The prepared datasets were the input data for creating the classifier of fake and real news. We selected decision trees for classification. The evaluation of the success of two different methods of preparation was carried out because of the success of the created classifier. We found a suitable dataset pre-processing technique by morphological group analysis. This technique could be used for improving fake news classification.

**Keywords:** fake news identification; text mining; natural language processing; Part of speech tagging; morphological analysis

## 1. Introduction

Nowadays, the Internet is part of our daily lives, and, at the same time, it is one of the main sources of information for the users. However, because of social networks or media, we are facing various fake news over the entire Internet. In practice, the entire fake news model requires an extensive amount of time and a relevant elaborate dataset [1,2]. Nowadays, more and more new messages and articles are emerging over the Internet. The fake news is a phenomenon that relates to various topics, which is continuously read by many users. This effect is very favorable for those who wrote these fake news [3]. The entire process shows that creating fake news can be defined as a multi-step approach that involves accomplishing or modifying the basic content that others have created [4]. Currently, fake news is an important area because there are many explanations and theories why people believe in fake news, and there are also various approaches on how to detect them [4,5].

Our research is focused on the analysis of our own created datasets [6], which contains articles in the Slovak language. Articles have been assigned to two specific classes according to the source of the publisher. These classes are verified source articles and articles published by a publisher classified as a fake or misleading content publisher. We analyzed these articles using text mining

methods. We focused on the linguistic side of the text using the morphological analysis of the content. Morphological analysis is an essential tool of how to explore the natural language. It contains the generic words and the shape characteristics of the term in context. The result is a set of tags that describes the grammatical categories of a given shape, especially the basic shape (lemma) and word pattern.

The aim of our paper is to find how to classify fake news messages using part of speech characteristics. According to our previous research, statistically significant differences were examined between the used parts of speech in fake and real news articles. The fake and real news were analyzed from the existing datasets of news and articles. In the aforementioned explorations of the authors, the news and articles were written in English. All those previous findings were proven mainly on publicly available datasets. In contrast, in the following research, we focused on articles in the Slovak language, which belongs to the flective languages. For flective languages, the expression of grammatical values is the leading role played by flexion (ending). The parts of speech and shape characteristics of words are especially important in these flective languages. In this article, we create a classifier for the fake and real news, and we try to find how to improve the classification through the appropriate data preparation. As a classifier, we practically use one of the simplest methods for classification, decision trees. In this article, we focus on possible classifications by creating groups with different morphological characteristics.

To accomplish the objective of the article, i.e. finding how to classify fake news messages using the part of speech characteristics, we follow the following procedure:

1.  We create our own datasets from real messages and from fake or misleading content. Datasets are created from messages in Slovak language.
2.  Datasets are pre-processed by means of the usage of basic techniques for natural language processing.
3.  To the words of examined dataset reports two different data preparation approaches are applied:

    Approach 1: Morphological tags are identified.
    Approach 2: According to our method (described below in our article), new categories are created from morphological tags.

4.  Decision trees are generated from the preprocessed data (Approaches 1 and 2).
5.  The basic characteristics of the created decision trees are compared with the evaluation of data preparation method, which provides better results. It is determined which method is more suitable for the Slovak language, as a flective type of language.

The article is divided into the following sections. We summarize the current status of fake news identification research field in Section 2. We describe the used datasets and methods of preparation of these datasets for further analysis in Section 3. Section 4 is the most important, which focuses on the description of the results of the analyses. We present the entire decision tree creation process and different views of the resulting decision tree model in this section. The discussion and conclusion are presented in the last section of this article.

## 2. Related Work

Basically, fake news is based on untrue information, which can be disseminated across the entire Internet. Fake news can also be interpreted as the publication of false statements [7], which are formed from non-existent news, hoaxes, and sensationalism of articles through social media. All around the world, we can detect fake news in various areas such as politics, education, and financial markets.

According to the authors [8], there are many approaches and suggestions for how to identify fake news and articles containing misinformation. Another approach to fake news detection assumes establishing a dataset that contains the concrete list of real and fake news. The extracted datasets must be analyzed and visualized in an appropriate form. The data collection is a very sophisticated and

complex area of fake news identification [9]. The data acquisition is one of the biggest challenges in the entire automated fake news detection process [10,11]. The main adversity is to find and verify suitable data, which are nearly related to the fake news issues [12].

The data acquisition is one of the biggest challenges in the entire automated fake news detection process [8,12]. The main adversity is to find and verify suitable data, which is nearly related to the fake news issues [13]. Mainly the fake news classification process is provided over a chosen specific and previously prepared dataset. Similarly, the authors of [14] used manually collected datasets for the fake news detection. The work in [13] provides a concept that can identify fake content using the automated system. The fake news detection is provided by means of the feature extraction. The fake news should also be identified due to social media, where there is a high possibility that misinformation is published. The authors of [15] used a chatbot, which is suitable to classify posts from social media. This approach was tested over Italian news classification, which provides more occasions to make the solution suitable for other languages. The weaknesses of the described concepts [16] is a small or few events related datasets. Considering the research results [12], it is possible to demonstrate how important is the effectiveness of the fake news detection, where the fake news credibility is automatically introduced.

As a result, we can detect fake news using text mining methods and artificial intelligence algorithms [17,18]. According to the authors of [19], we can divide the detection methods into linguistic, deceptive, or predictive modeling and clustering, and then we clarify content cue and non-text based methods. They applied algorithms such as decision trees, neural networks, and naive Bayes classifiers. The authors of this research applied a deep neural network, which can identify the fake news articles and subjects in the network. Statistically significant differences were examined between the used parts of speech in fake and real news articles [20,21].

Different approaches to the detection of fake news have been revealed by many authors [21,22], as a possibility for how to detect fake news by means of machine learning [23]. Subsequently, in research [15], the determination between the fake and the real news was proven. In practice, it is often desirable if the datasets are analyzed and visualized in the appropriate form.

## 3. Methodology

It was necessary to create an experimental dataset to get as accurate results as possible. Usable datasets for the Slovak language are not yet available. It was, therefore, necessary to create our own dataset. However, we wanted to avoid strict article labeling and our subjective classifications. For this reason, we used the rating of web portals from www.konspiratori.sk (conspirators), which is devoted to the identification of suspicious webpages and articles. This web service is a public database of websites that provide dubious, deceptive, fraudulent, conspiratorial, or propaganda content. The methodology and procedures, which describe how and why the portal is finally included in the database of the possible conspiracy websites, can be found at [24].

We used articles that contain the keywords "NATO" and "Russia" for dataset creation. We assumed that these keywords would have a significant difference in presentation and reporting on portals that deploy against the traditionally established newspaper portals. The created dataset contains 160 articles. We assigned fake or real information to each article, according to the newspaper publisher, in which the article was published. Such a consideration encounters several problems. Published articles on a portal can be fake or misleading. Such portals publish several real "undistorted" articles. Sometimes a serious portal can also "publish" an article with misleading content. Our aim was to avoid our own subjective opinion and to rely only on the available information, which was available for the newspaper portal. Our entire dataset was then divided into 80 articles of fake type (according to the publisher of the article, which is included in the database konspiratori.sk) and 80 articles of type real (verified publisher) (Figure 1).

| | content | publisher | key_word | label |
|---|---|---|---|---|
| 1 | V médiách sa objavila informácia, že počas dvo... | http://me...s.sk/ | NATO | fake |
| 2 | Na základe posledných udalostí sa nový ministe... | http://me...s.sk/ | NATO | fake |
| | ... | ... | ... | ... |
| 159 | V júli ťažba dosahovala 11,15 milióna barelov ... | https://h...e.sk/ | Rusko | real |
| 160 | Premiér uviedol, že prechod na štvordňový prac... | https://h...e.sk/ | Rusko | real |

**Figure 1.** Preview of the created dataset (publisher of the articles is anonymized).

The Slovak language has complex rules for word inflection. The most important contribution to the automated morphological analysis of the Slovak language was a proposal of the morphologically annotated corpus of the by Slovak National Corpus. The corpus defines a set of Part-of-speech (POS) tags and annotation guidelines for the Slovak language, and it is still the most used set of morphological tags [25]. The tags are of unequal length, but most tags follow the same structure for the same inflectional paradigm. The information of the part of speech is often more important than the structure or the length of the tags [26,27] and it is encoded in the first position of every tag. The second position usually marks an inflectional paradigm for words that have this category. The code for the position repeats that of the corresponding parts of speech, e.g., "SS..." stands for a noun with a noun-like inflectional paradigm, and "PS..." for a pronoun with a noun-like inflectional paradigm.

We present noun and pronoun as an example of the positions in the morphological tags for each part of speech categories in Table 1.

**Table 1.** Morphological division of the parts of speech of the explored dataset.

| | Noun | | Pronoun | |
|---|---|---|---|---|
| Position | Possible Values | Description | Possible Values | Description |
| 1 | S | part of speech tag | P | part of speech tag |
| 2 | SAFU | paradigm | SAFU | paradigm |
| 3 | mifn | gender | mifn | gender |
| 4 | sp | number | sp | number |
| 5 | 1234567 | case | 1234567 | case |

For example, the morphological tag "SSms1" means: S (part of speech tag—"Noun"); S (paradigm—"substantive"); m (gender—"man"); s (number—"simple"); 1 (case—"nominative"). A detailed description of Slovak Morphosyntactic Tagset can be found in [15].

Morphological analysis was applied to all articles from the dataset, which was created by us. So-called morphological tags were assigned to each word of the articles. This was done using the tool TreeTagger [28]. TreeTagger is a tool for the part of speech annotation and lemma information. It was developed by Helmut Schmid in the TC project at the Institute for Computational Linguistics of the University of Stuttgart [29].

The application of our methods is based on the average representation of a morphological tag or morphological group in fake vs. non-fake reports. For this purpose, it was also necessary to calculate the number of words in each article. The final dataset with morphological tags and the number of words in each article is shown in Figure 2.

| | content | publisher | key_word | label | tags | word_count |
|---|---|---|---|---|---|---|
| 1 | V médiách sa objavila informácia, že počas dvo... | http://████s.sk/ | NATO | fake | Eu6 SSnp6 R VLdscf+ SSfs1 Z O Eu2 NNip2 SSip2 ... | 935 |
| 2 | Na základe posledných udalostí sa nový ministe... | http://████s.sk/ | NATO | fake | Eu6 SSis6 AAfp2x SSfp2 R AAms1x SSms1 AAfp2x S... | 695 |
| | ... | ... | ... | ... | .... | .... |
| 159 | V júli ťažba dosahovala 11,15 milióna barelov ... | https://████e.sk/ | Rusko | real | Eu6 SSis6 SSfs1 VLescf+ 0 Z 0 NSis2 SSip2 Dx S... | 317 |
| 160 | Premiér uviedol, že prechod na štvordňový prac... | https://████e.sk/ | Rusko | real | SSms1 VLdscm+ Z O SSis1 Eu4 AAis4x AAis4x SSis... | 201 |

**Figure 2.** Preview of the dataset created after identifying morphological tags and calculating the number of words in each article (publisher of the articles is anonymized).

The number of unique morphological tags identified in the dataset of articles was 799. Subsequently, the individual tags were grouped, and the relative abundance of groups of individual tags in the analyzed articles was calculated. The grouping was carried out in two different approaches.

Approach 1: The first "simpler" groups were created based on the similarity of the morphological tag, which belongs to the so-called "parts of speech". Slovak morphological tags have a fixed order of characters, with each character in a morphological tag expressing a morphological property. The first character in each tag expresses the part of speech categories (noun, adjective, pronoun, numeral, participle, etc.). Therefore, the tags were grouped according to these parts of each speech categories (the number of such created groups was 21) and the relative abundance of these groups in the examined articles were calculated (Figure 3).

| | content | label | tags | word_count | tags_group | D | Q | 0 | V | G | ... | S | R | T | O | Y |
|---|---|---|---|---|---|---|---|---|---|---|---|---|---|---|---|---|
| 1 | V médiách sa objavila informácia, že počas dvo... | fake | Eu6 SSnp6 R VLdscf+ SSfs1 Z O Eu2 NNip2 SSip2 ... | 935 | ESRVSZ OENSE SESRE SVSSS0 Z... | 0.014957 | 0.002137 | 0.017094 | 0.110043 | 0.027778 | ... | 0.240385 | 0.016026 | 0.024573 | 0.075855 | 0.003205 |
| 2 | Na základe posledných udalosti sa nový ministe... | fake | Eu6 SSis6 AAfp2x SSfp2 R AAms1x SSms1 AAfp2x S... | 695 | ESASRA SAS%% ESE%% VZOSVV E... | 0.035920 | 0.000000 | 0.007184 | 0.125000 | 0.017241 | ... | 0.278736 | 0.011494 | 0.025862 | 0.073276 | 0.002874 |
| 3 | Vláda Mikuláša Dzurindu na utajenom zasadnutí ... | fake | SSfs1 SSms2:r SSms2:r Eu6 Gtns6x SSns6 AAis2x... | 565 | SSSEG SAS0VS ASWEP SZORD VV... | 0.031802 | 0.000000 | 0.010601 | 0.104240 | 0.010601 | ... | 0.303887 | 0.015901 | 0.033569 | 0.065371 | 0.001767 |

**Figure 3.** Preview of the dataset (Approach 1) created after identifying morphological tags and calculating the number of words in each article.

Approach 2: The second approach to grouping used fixed positions in morphological tags. Groups were created as pairs, combining the first character (part of speech tag) with the other characters in the tag (other characteristics). Thus, the morphological tag "SSms1" was assigned to the following groups: SS (part of speech [1.char in tag]—paradigm [2.char in tag]), S_m (part of speech [1.char in tag]—gender [2.char in tag]), S_s (part of speech [1.char in tag]—number [3.char in tag]), S_1 (part of speech [1char in tag]—case [4.char in tag]). The previously created groups were formed and presented as an improvement (refinement) of the groups, according to the part of speech with respect to other morphological characteristics. As a result, 158 groups were created. The relative abundance of these groups in the examined articles were calculated (Figure 4).

| | content | label | tags | word_count | A__2 | Gk | E_3 | VL | S_i | S__6 | ... | S_f | S___: | G_i | AA | VM |
|---|---------|-------|------|-----------|------|----|----|----|----|------|-----|-----|-------|-----|----|----|
| 1 | V médiách sa objavila informácia, že počas dvo... | fake | Eu6 SSnp6 R VLdscf+ SSfs1 Z O Eu2 NNip2 SSip2 ... | 935 | 0.027778 | 0.003205 | 0.011752 | 0.032051 | 0.076923 | 0.038462 | ... | 0.100427 | 0.032051 | 0.011752 | 0.082265 | 0.003205 |
| 2 | Na základe posledných udalosti sa nový ministe... | fake | Eu6 SSis6 AAfp2x SSfp2 R AAms1x SSms1 AAfp2x S... | 695 | 0.022989 | 0.004310 | 0.012931 | 0.031609 | 0.076149 | 0.064655 | ... | 0.106322 | 0.068966 | 0.002874 | 0.091954 | 0.000000 |
| 3 | Vláda Mikuláša Dzurindu na utajenom zasadnutí ... | fake | SSfs1 SSms2:r SSms2:r Eu6 Gtns6x SSns6 AAis2x:... | 565 | 0.040636 | 0.001767 | 0.005300 | 0.053004 | 0.083039 | 0.031802 | ... | 0.118375 | 0.061837 | 0.003534 | 0.116608 | 0.000000 |

**Figure 4.** Preview of the dataset (Approach 2) created after identifying morphological tags and calculating the number of words in each article.

Both pre-processed datasets (using Approaches 1 and 2) were used as the input to the creation of decision trees for classification fake/real news. The main aim of this step of the applied methodology was to verify how feasible is the morphological analysis for the successful classification of fake or real news. Data classification is one of the data mining techniques used to extract models describing important data classes. Some of the common classification methods used in data mining are decision tree classifiers, Bayesian classifiers, k-nearest-neighbor classifiers, case-based reasoning, genetic algorithms, rough sets, and fuzzy logic techniques. Among these classification algorithms, decision tree algorithms are the most commonly used because they are easy to understand and cheap to implement.

The basic idea behind any decision tree algorithm is as follows:

1. Select the best attribute using attribute selection measures (ASM) to split the records.
2. Make that attribute a decision node and break the dataset into smaller subsets.
3. Start tree building by repeating this process recursively for each child until one of the conditions is matched:

   - All tuples belong to the same attribute value.
   - There are no more remaining attributes.
   - There are no more instances.

ASM is a heuristic method for selecting the splitting criteria that partition data into the best possible manner. It is also known as splitting rule because it helps to determine breakpoints for tuples on a given node. ASM provides a rank to each feature (or attribute) by explaining the given dataset. The attribute with the best score will be selected as a splitting attribute (source). In the case of a continuous-valued attribute, split points for branches also need to be defined. The most popular selection measures are information gain, gain ratio, and Gini index [21,30].

## 4. Results

We created several decision trees to verify the suitability of using morphological tags. They differed in the type of input data for the classification. Subsequently, we compared the accuracy of all the decision trees methods. We experimented with a selection of measures and tree depth. The accuracy was calculated for each created tree individually using the following equation:

$$Accuracy = \frac{Number\ of\ correct\ predictions}{Total\ number\ of\ made\ predictions} \tag{1}$$

Accuracy is the ratio of the number of correct predictions to the total number of input samples. It works well only if there are an equal number of samples belonging to each class. This condition was met in the examined data. The analyzed dataset was divided into the ratio 70:30, i.e., 70% of randomly selected records were used to create decision trees. We created decision trees by setting different depths of a tree using entropy as a selection measure.

Results for Approach 1: Figure 5 shows the calculated accuracy for the used decision trees. There were 10 decision trees created, with a fixed maximum depth. Twenty-one properties were recorded in the input file, which was used to classify the fake and real groups of news. The highest reached accuracy was 0.6875. This accuracy was observed in a tree with a maximum depth of 7.

```
Tree#1   Max Deep: 3        Accuracy:0.6458333333333334
Tree#2   Max Deep: 4        Accuracy:0.625
Tree#3   Max Deep: 5        Accuracy:0.625
Tree#4   Max Deep: 6        Accuracy:0.5625
Tree#5   Max Deep: 7        Accuracy:0.6875
Tree#6   Max Deep: 8        Accuracy:0.5833333333333334
Tree#7   Max Deep: 9        Accuracy:0.5625
Tree#8   Max Deep: 10       Accuracy:0.5833333333333334
Tree#9   Max Deep: 11       Accuracy:0.6666666666666666
Tree#10  Max Deep: 12       Accuracy:0.5833333333333334
```

**Figure 5.** Calculated accuracy of decision trees created for different maximum depths from a dataset created according to Approach 1.

Result for Approach 2: An improved preparation of the input dataset was described in Approach 2. The input into the decision tree algorithm had 158 features, which represented the relative occurrence of created groups of morphological tags in the examined articles. These features were used to classify articles into fake and real newsgroups. As a result (Figure 6), it is evident that, by using the features or groups of morphological tags that are created in this approach, we received a better classification accuracy, while the best-observed accuracy was 0.75 for a tree with a maximum depth of 9.

```
Tree#1   Max Deep: 3        Accuracy:0.5833333333333334
Tree#2   Max Deep: 4        Accuracy:0.6458333333333334
Tree#3   Max Deep: 5        Accuracy:0.6666666666666666
Tree#4   Max Deep: 6        Accuracy:0.6041666666666666
Tree#5   Max Deep: 7        Accuracy:0.7291666666666666
Tree#6   Max Deep: 8        Accuracy:0.7083333333333334
Tree#7   Max Deep: 9        Accuracy:0.75
Tree#8   Max Deep: 10       Accuracy:0.6666666666666666
Tree#9   Max Deep: 11       Accuracy:0.7083333333333334
Tree#10  Max Deep: 12       Accuracy:0.7083333333333334
```

**Figure 6.** Calculated accuracy of decision trees created for different maximum depths from created dataset pre-processed in means of the Approach 2.

We introduce an example of creating a decision tree with the best classification accuracy for completeness.

In the decision tree example, each internal node has a decision rule that splits the data. In Figure 7, each node contains five important pieces of information about itself. The first piece of information (row) is the most important condition for the division. For example, in the first node, the condition is "Eu $\leq$ 0.093". This means that whether the relative abundance of the Eu group (derived from morphological tags) is less than a given number is a condition. The second piece of information in the node represents the Gini index value found by the Gini method to create split points. The third piece

of information is the number of examples (from the training set) that includes the node (112 for the first node), which are divided into 57 fake and 55 real messages (fourth piece of information about node), and this node has a majority fake group (the last piece of information). It should be noted that, although we worked with 160 fake/real articles, the tree in the sample was made of 112 news articles. The rest were used to validate the accuracy of the tree as test cases.

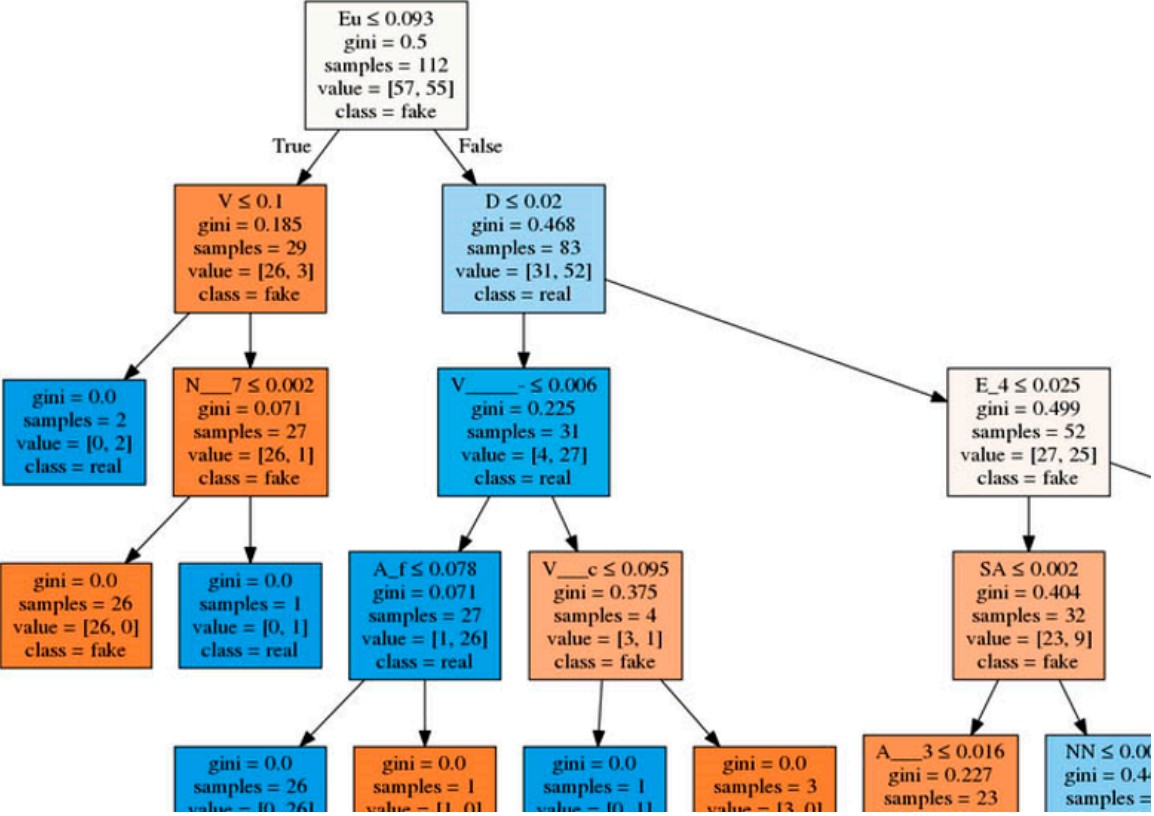

**Figure 7.** Sample part of the created decision tree.

## 5. Discussion

Despite the noticeable improvement in the classification, the accuracy of the decision trees is not high. We assume this can be caused by a small number of examples in the created dataset. The 160 articles from the randomly examined dataset divided into a training and test set at a ratio of 70:30 are not enough for the generalization purpose. The decision trees were generated from 112 records, and the remaining 48 records were used to calculate the accuracy. However, decision trees as a typical machine learning algorithm need many input dataset entries to create them. We expect to add more articles to the dataset in the future.

The lower accuracy of created decision trees can also be explained by the nature of the articles in the dataset. The dataset was constructed according to the fact that the publisher of articles was or was not included in the database available at [24]. The designation of an article as fake means that it has been published on portals that are well-known publishers of fake, misinformation, or misleading articles. However, this does not mean that any published article on such portals must also be fake or misleading. Often such news portals publish real and truthful articles in order to increase their credibility. It is a common practice that misleading or false information is added to a real article. These facts probably also influenced the quality of the input dataset.

## 6. Conclusions

Decision tree classification is one of the most basic and practically the simplest methods of machine learning. If other (more appropriate) methods were used, e.g., neural networks, for the created datasets, the accuracy of the classification would increase. However, we deliberately chose decision trees because of the easy interpretation of their results. The decision trees that were created will be analyzed by media experts to create a description of the concepts from them, i.e. they will create of a set of rules to extract (from decision trees) and describe the morphological characteristics that are typical for misleading or fake news.

**Author Contributions:** Conceptualization, J.K.; Methodology, J.K.; Project administration, J.K.; Resources, J.O.; Validation, J.K. and J.O.; Visualization, J.O.; Writing—original draft, J.K.; and Writing—review and editing, J.O. All authors have read and agreed to the published version of the manuscript.

**Acknowledgments:** This work was supported by the Operational Program: Research and Innovation—project "Fake news on the Internet—identification, content analysis, emotions", co-funded by the European Regional Development Fund under contract 313011T527.

**Conflicts of Interest:** The author declares no conflict of interests.

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
