# Peer review of "Improvement of Misleading and Fake News Classification for Flective Languages by Morphological Group Analysis"

_informatics, doi:10.3390/informatics7010004_

Round 1

Reviewer 1 Report

This is an interesting paper about an important topic with some new results achieved. However, some improvements would be beneficial. Therefore some my suggestions are in the attached file, take them into account, please.

Author Response

Please kindly find authors' reply in the attachment.

Reviewer 2 Report

The article presents methods for fake news detection (slovak language) based on decision trees under development by authors. It presents the model and partial results and possibilities of better efficiency of the presented approach. Although authors state it to be rather position paper than complete and final research, I am sure it would be interesting paper showing their progress. Maybe some more detailed examples of efficiency on more extensive data sets (from different cases) could raise the significance of the paper. Also comparisons with another concurrent works may be of interest for potential readers.

Nevertheless I support the publication of the paper considering above mentioned remarks.

Author Response

(The authors gave the same response as above.)

Reviewer 3 Report

Overall evaluation: The paper is a bit confuse in the sense that it does not follow a clear reasoning line. Several subjects merge into and evade from the main idea. A higher terminology rigor may help with this problem. The question that holds is, that giving you state in section 5 that the sample is too low to be conclusive, why don't you raise the sample after a proper threshold before publishing this paper?

From my perspective, the title does not properly reflect the paper. First of all, the sample is small (160 units) and narrow (NATO and Russia). Perhaps the words like "exploration", "initial" or "small" should be included. Also, PoS tagging is not a novel approach for machine learning or decision tree, I am not familiar with research in Slovak language nor in Slovak fake-news, but it would be advisable to these words also to appear in the title. The results, of .68 and .75, should appear in the abstract.

Fake-news as a clickbait approach can be considered as the least concern problem, therefore I do not believe that to be its "main-goal". I could not either derive this idea from reference [3], maybe if you change the word "click" for "spread" may suit betters. Assertions about reference [4] also do not feel right. My first suggestion is to carefully review the bibliography discussed concepts in order to present them with higher accuracy.

You talk about previous research of yours that is used to support this one but no citation for a paper is made. Why is it? Then you assume that the same result of that research suits for the Slovak language used in this paper, yet the Slovak language is presented as different from the English. Maybe a full explanation may cover these gaps.

About the assertion "The identification and the classification of fake news are based on a collected and previously prepared dataset." This may be true for a certain type of approach but not a universal truth. May you refine this assertion for properly present your belief?

I believe that a comprehensive introduction to the Slovak language could worth for people researching other languages that may find similarities. Also, some details about the implementation (what programming language and APIs used) and a link for the data set and the program would be worthful.

Author Response

(The authors gave the same response as above.)

Round 2

Reviewer 1 Report

The improved version of the paper seems to be in accord with the reviewers' requirements. Just check please the newly introduced sentences for spelling mistakes, there are some.